# Socioeconomic Status and Prosocial Behavior: The Mediating Roles of Community Identity and Perceived Control

**DOI:** 10.3390/ijerph181910308

**Published:** 2021-09-30

**Authors:** Yanli Wang, Chao Yang, Yanchi Zhang, Xiaoyong Hu

**Affiliations:** 1School of Education Science, Jiangsu Normal University, Xuzhou 221116, China; wyl1054616437@163.com; 2School of Psychology, Guizhou Normal University, Guiyang 550025, China; yangchaopsy632@163.com; 3School of Psychology, Northwest Normal University, Lanzhou 730070, China; zhangycpsy@163.com; 4Faculty of Psychology, Southwest University, Chongqing 400715, China

**Keywords:** socioeconomic status, prosocial behavior, community identity, perceived control, serial mediation

## Abstract

Background: Previous studies have examined the association between socioeconomic status and prosocial behavior, but the underlying mechanism between them is unclear. The present study aimed to examine the serially mediating roles of community identity and perceived control in this relationship. Methods: Using the convenient sampling technique, a total of 477 Chinese adults from Chinese communities, and ranging in age from 20 to 65 completed the questionnaires for objective socioeconomic status, the MacArthur scale of subjective socioeconomic status, an eight-item community identity scale, the perceived control scale, and a prosocial tendencies measure. Bivariate correlation analysis and regression analysis were used to examine the relationships among the major variables. Results: Socioeconomic status was positively associated with prosocial behavior. It was also found that community identity and perceived control played mediating roles between socioeconomic status and prosocial behavior, respectively. In addition, community identity and perceived control also had a serially mediating role in the relationship. Conclusions: Community identity and perceived control played a serially mediating role in the association between socioeconomic status and prosocial behavior. The findings in the present study contribute to understanding the underlying mechanism in the association between socioeconomic status and prosocial behavior among adults, and have important implications for interventions aimed at improving prosocial behavior in lower-status individuals.

## 1. Introduction

Urban and rural communities are the basic units of social governance, which aims to form a good community atmosphere of friendship, partnership, and mutual help. Prosocial behavior refers to voluntary behavior that is beneficial to others or society, including cooperation, help, comfort, and donation [1]. It can improve individuals’ happiness [2,3], perceptions of meaning in life [4], and relieve depression and anxiety [5]. Therefore, it is necessary to explore prosocial behavior in the context of the community.

Socioeconomic status is a social classification used to reflect the relative position of individuals on the social ladder [6]. Both animal and human studies have proven that socioeconomic status affects all aspects of individuals’ lives [7,8,9], including prosocial behavior. For example, low-status people were more likely to identify with social values oriented toward egalitarianism and the well-being of others [10], and showed higher levels of empathy [11]. Interestingly, the association between socioeconomic status and prosocial behavior remains controversial. On the one hand, cost–benefit analyses [12] showed that prosocial behavior consumed individuals’ resources, including time and effort, while increasing danger, embarrassment, and disruption of ongoing activities. With the increase of resources consumed, individuals exhibited less prosocial behavior. Thus, one researcher [13] found that prosocial behavior consumed more time and money among lower-status individuals and they had less prosocial behavior. Numerous empirical studies also support this result. For example, Andreoni, Nikiforakis, and Stoop [14] found that the poor were less likely to return misdelivered envelopes, even if it was not good for them to keep them. A series of studies conducted by Korndörfer et al. [13] also found that people with lower status spent a lower percentage of their income on philanthropy and were less likely to make charitable donations than those with higher status. In addition, they were less friendly in daily communication and less reliable when communicating with strangers in trust games. Moreover, other researchers [15] using Chinese community residents also found that there was a significant positive correlation between socioeconomic status and altruistic behavior. On the other hand, from the perspective of social cognition of socioeconomic status [6], this indicated that lower-status people had less material resources and faced more uncertainty and unpredictability, which led them to form a social cognitive tendency of contextualism. On the contrary, upper-status people had more social resources, which led them to form a social cognitive tendency of solipsism. Different cognitive tendencies had an effect on prosocial behavior. Ample empirical studies found that lower socioeconomic status was related to more prosocial behavior [10,16]. Overall, the relationship between socioeconomic status and prosocial behavior is still controversial.

How does socioeconomic status affect prosocial behavior? Bronfenbrenner’s ecosystem theory [17,18] points out that a person’s development is affected, not only by personal characteristics, but also by microsystems. Moreover, socioeconomic status is a factor of the microsystem, and community identity is a personal characteristic variable. On the one hand, the social identity approach came from social identity theory [19,20], and the self-categorization theory [21] believes that groups can shape individual’s psychology by internalizing their sense of self, and pointed out that belonging to a group affects an individuals’ cognition, emotion, and behavior [22]. The social cure perspective, derived from the social identity approach, believes that group identity is the basis of a volunteers’ motivation to participate in voluntary activities [23], which could increase environmental protection behavior [24]. It is well-known that social identity can effectively predict prosocial behavior [25,26,27]. Community identity, as a special type of social identity [28,29], refers to the recognition, approval, and value of community members for the common values of the community, including emotional identity and functional identity. The former is embodied in whether residents care about other people’s views about their own community, whether they have special feelings for the community, and so on. The latter is reflected in the residents’ recognition of the convenience of the community, the level of management, the environmental conditions, and whether the community can meet the needs of the family [30]. Yang and Xin [29] explored the influence of community identity on positive intentions, in the context of time pressure, and found that people with higher community identity had a higher level of intention to help others than those with lower community identity. Fritsche et al. [24] summarized that community identity would increase individuals’ support for environmental protection behaviors. A recent study [31] also found that the intensity of people’s identity with the local community had a positive predictive effect on their willingness to participate in prosocial normative behaviors in the community. Researchers [32] conducted a two-wave longitudinal online survey during the COVID-19 pandemic, and found volunteer role identity at T1 (pre-pandemic) positively affected perceptions of volunteer–beneficiary intergroup closeness at T1, which also in turn affected community identity at T1. This, in turn, positively predicted COVID-19 collaborative assistance for T2 (3 months later). On the other hand, compared with those with upper status, lower-status people were more likely to live in an environment of higher crime rates, parental conflict, and community violence, which was not conducive to the formation of community identity [33]. Furthermore, some empirical studies [15,34] have pointed out that there was a significant positive correlation between socioeconomic status and community identity. Moreover, Wang et al. [34] found community identity played a mediating role between socioeconomic status and altruistic behavior.

Perceived control is the individual’s perception of his or her ability to control events and the expectation of consistency between his or her behavior and the outcome of events [35]. On the one hand, a meta-analysis [36] found that there was a significant and positive correlation between perceived control and prosocial behavior. Morris, Sim, and Girotto [37] also found that people with lower perceived control had lower expectations of their opponents’ cooperation than those with higher perceived control. On the other hand, lower-status people had less access to education, poor living conditions, and often faced the threat of unemployment, which was not conducive to the pursuit of their goals and reduced their perceived control [6,38]. Moreover, a previous study [39] found that control beliefs had a mediating role between socioeconomic status and the exercising of behaviors or intentions.

In addition, socioeconomic status affects prosocial behavior, not only through community identity or perceived control, but also through the serially mediating role of the two. A group-based control restoration model considered that individuals could gain control through groups. In other words, people could explain themselves by identifying with the group in order to maintain their perceived control [40,41]. Greenaway et al. [42] found that social identity affected individual’s mental health through perceived control. Social identity can also promote well-being by meeting global psychological needs, including feelings of belonging, self-esteem, perceived control, and sense of meaning [43].

To sum up, although ample studies have identified an association between socioeconomic status and prosocial behavior, they have not paid sufficient attention to the underlying mechanism of the link. Here, we expand on previous studies, in several respects. First, although Wang et al. [34] indicated that community identity mediated the association between socioeconomic status and altruistic behavior, it is unclear whether community identity mediates the connection of socioeconomic status with prosocial behavior. Second, no study has directly explored the mediating role of perceived control in the relation between socioeconomic status and prosocial behavior. Third, a group-based control restoration model indicated that social identity affected perceived control [40,41], and perceived control mediated the association between social identity and mental health [42]. Thus, based on theoretical and empirical studies, the present study aimed to explore the association between socioeconomic status and prosocial behavior, and the serially mediating role of community identity and perceived control in this association. Four hypotheses were proposed: (1) socioeconomic status is associated with prosocial behavior; (2) community identity mediates the association between socioeconomic status and prosocial behavior; (3) perceived control mediates the association between socioeconomic status; and (4) community identity and perceived control play a serially mediating role between socioeconomic status and prosocial behavior. The study model is presented in Figure 1.

## 2. Materials and Methods

### 2.1. Participants

Using the convenient sampling technique, 477 adults completed valid questionnaires (out of the 560 who were approached to participate), excluding incomplete answers, random answers, and an obvious tendency of answers. Their age ranged from 20 to 65, with an average age of 30.63 years (SD = 8.03). Full descriptive statistics of the samples are presented in Table 1.

### 2.2. Measures

#### 2.2.1. Socioeconomic Status

According to the existing research [44], socioeconomic status was calculated through adding the standard scores of subjective and objective indicators. Among them, objective socioeconomic status was measured by education level, occupation, and monthly income. With reference to previous studies [34,45], education level was divided into six categories: “primary school and below”, “junior middle school”, “senior high school”, “associate degree”, “college graduate”, and “master and above”, which were assigned a score of 1–6, respectively. Occupation (reverse scoring) was divided into state and social managers, managers, private entrepreneurs, professional and technical personnel, office workers, individual industrial and commercial households, business and service employees, industrial workers, agricultural laborers, urban and rural unemployed, and semi-unemployed, which was based on cultural resources, economic resources, and organizational resources [46,47]. Monthly income was divided into ten levels, including “less than 1000 yuan”, “1001–3000 yuan”, “3001–5000 yuan”, “5001–7000 yuan”, “7001–10,000 yuan”, “10,001–15,000 yuan”, “15,001–30,000 yuan”, “30,001–50,000 yuan”, “50,001–100,000 yuan”, and “more than 100,001 yuan” [48]. Referring to the existing studies [34,49], the three indicators were standardized, and a principal component analysis was carried out to generate an eigenvalue greater than 1, which explained 60.99% of the variance. We obtained a comprehensive formula for calculating objective socioeconomic status: objective socioeconomic status = (0.797 × Z_education_ + 0.760 × Z_monthly income_ + 0.785 × Z_occupation_)/1.830. The factor loadings of the three indicators were 0.797, 0.760, and 0.785, respectively, and the eigenvalue of the first factor was 1.830. The higher the score, the higher the objective socioeconomic status. Subjective socioeconomic status was measured using the MacArthur scale of subjective socioeconomic status [50]. The scale has commonly been used in previous studies [51,52,53]. We presented the participants with a 10-step ladder to reflect the different socioeconomic statuses of the Chinese people. Participants were asked to imagine that the 10-step reflected the different statuses of the Chinese people. Individuals at the top of the ladder had the best living situation, the best education, and the highest income. While individuals at the bottom of the ladder lived in the worst situation, received a basic education, and had the lowest income. Finally, participants were asked to evaluate their position in the ladder, according to their own situation (such as income, education level, and occupation).

#### 2.2.2. Community Identity

The community identity scale, developed by Xin and Ling [30], was used to assess community identity. It consisted of 8 items. Two sample items were: “compared with other places, the environmental conditions of the community here are satisfactory” and “living in this community is very convenient”. Each item was answered on a 6-point Likert scale ranging from 1 = not true at all, to 6 = absolutely true, with higher scores indicating higher community identity. The scale was found to have high internal consistency, with a Cronbach’s alpha of 0.91 in the original application [30]. The Cronbach’s alpha for the scale in the study was 0.91.

#### 2.2.3. Perceived Control

The perceived control scale, developed by Lanchman and Weaver [54], was used to examine perceived control. There were 12 items. A sample item was “if I really want to do something, I can usually find a way to succeed.” Each item was answered on a 7-point Likert scale ranging from 1 = totally disagree, to 7 = totally agree, with higher scores indicating higher perceived control. The scale has been used with Chinese samples and was found to have good construct validity and satisfactory internal consistency (α = 0.79) [55]. In this study, the Cronbach’s alpha of the scale was 0.83.

#### 2.2.4. Prosocial Behavior

The prosocial tendencies measure, developed by Carlo and Randall [56], was conducted to assess prosocial behavior. It consisted of 26 items. A sample item was: “it is not difficult for me to provide help to someone when they are in a terrible and desperate need”. Each item was answered on a 5-point Likert scale ranging from 1 = does not describe me at all, to 5 = describes me greatly, with higher scores indicating more prosocial behavior. The scale has been used with Chinese samples and was found to have good construct validity and satisfactory internal consistency (ranging from α = 0.56 to α = 0.79) [57]. In this study, the Cronbach’s alpha of the scale was 0.93.

### 2.3. Statistical Analysis

SPSS22.0, PROCESS, and AMOS22.0 software were used to examine our hypothesis. Before conducting analysis, skew and kurtosis were carried out to examine the assumptions of normality and homoscedasticity. The results showed that socioeconomic status, community identity, perceived control, and prosocial behavior were normally distributed, with skew and kurtosis values within normal limits (skew < +/−2, kurtosis < +/−7) [58]. First, the reliability and validity of the measurement items were checked. Second, the common method bias was checked by procedural control and statistical control [59]. Third, bivariate correlation analysis was used to examine the relationships among the major variables. Fourth, we tested the mediated model using the PROCESS macro (http://www.afhayes.com, accessed on 11 April 2012) for SPSS [60], with 5000 iterations of computing samples and bias-corrected 95% confidence intervals. If the confidence interval of 95% did not include zero, the result was significant at a *p* < 0.05 level.

## 3. Results

### 3.1. Validity and Reliability Analysis

Validity and reliability analyses were conducted. First, average variance extracted (AVE) and composite reliability (CR) were used to examine convergent validity. Results showed that the AVE values were all greater than 0.45 [61] and the CR values were all above 0.70 [62], which suggests the acceptable convergent validity of the constructs. Second, discriminant validity was verified using a Fornell–Larcker test, that is, whether the square root of AVE of each construct was higher than the correlation coefficients with other constructs. Results showed that the square root of each construct AVE was greater than its correlation with other constructs. Third, confirmatory factor analysis was conducted to evaluate the suitability of the research model. Results showed that the research model was acceptable (*χ*^2^/*df* = 2.96, *GFI* = 0.96, *IFI* = 0.97, *CFI* = 0.97, *RMSEA* = 0.06). Table 1 shows the results of our analysis.

### 3.2. Preliminary Analysis

Owing to the use of the questionnaire method to collect data in the study, there may have been the common method bias. This was checked through procedural control and statistical control [59]. The former emphasized anonymity, confidentiality, and the use of data only for scientific research, and the latter was conducted through Harman’s single factor test. Results showed that the characteristic roots of eleven factors exceeded 1, and the variance explained by the first factor was 22.93%, which was far less than the 40% of the critical judgment criteria. Therefore, there was not a serious common method bias in the study.

The means, standard deviations, and inter-correlation coefficients for the major variables are presented in Table 2. The results showed that socioeconomic status was positively associated with community identity (*r* = 0.27, *p* < 0.01), perceived control (*r* = 0.20, *p* < 0.01), and prosocial behavior (*r* = 0.18, *p* < 0.01). Community identity was positively associated with perceived control (*r* = 0.22, *p* < 0.01) and prosocial behavior (*r* = 0.43, *p* < 0.01). While perceived control was positively related to prosocial behavior (*r* = 0.19, *p* < 0.01). Therefore, it was suitable for mediation effects.

### 3.3. Mediating Effects

After standardizing all variables, socioeconomic status was taken as an independent variable, prosocial behavior as a dependent variable, and community identity and perceived control as mediating variables. Using PROCESS software for SPSS22.0 (Model 6) [60], the present study examined the serially mediating role of community identity and perceived control between socioeconomic status and prosocial behavior. As shown in Figure 2, the results showed that socioeconomic status was positively related to prosocial behavior (c) and community identity (a_1_); both socioeconomic status (a_2_) and community identity (d) were positively related to perceived control; meanwhile, community identity (b_1_) and perceived control (b_2_) were both positively related to prosocial behavior. However, socioeconomic status was not directly related to prosocial behavior (c’).

Moreover, the direct test of the mediating effect showed that the indirect effect of community identity and perceived control was significant (total mediating effect = 0.08, 95% CI (0.05, 0.11)). The mediating effect contained three mediating pathways, namely the independent mediating effect of community identity (mediating effect = 0.07, 95% CI (0.04, 0.09)); the serial mediating effect of community identity and perceived control (mediating effect = 0.01, 95% CI (0.01, 0.02)), and the independent mediating effect of perceived control (mediating effect = 0.01, 95% CI (0.01, 0.02)). However, the direct association between socioeconomic status and prosocial behavior was not significant (direct effect = 0.02, 95% (−0.03, 0.07)). The mediating model and the values of the pathway are presented in Table 3.

## 4. Discussion

This present study explored the association between socioeconomic status and prosocial behavior and the serially mediating role of community identity and perceived control. The results showed that there was a significant positive correlation between socioeconomic status and prosocial behavior. While, socioeconomic status affected prosocial behavior, not only through the independent mediating role of community identity or perceived control, but also through the serially mediating role of community identity and perceived control. However, when community identity and perceived control were added, the direct effect of socioeconomic status on prosocial behavior was not significant.

First, the correlation analysis found that there was a significant positive correlation between socioeconomic status and prosocial behavior, which was consistent with previous studies [13]. The reasons might be the following: on the one hand, lower-status people had scarce social resources, while prosocial behavior consumed their own resources, such as time and money [12]; on the other hand, lower-status people were more likely to face economic poverty and poor health, leading them to pay more attention to meeting immediate needs [63]. All these would reduce their prosocial behavior.

Second, this study also found that community identity played a mediating role between socioeconomic status and prosocial behavior, which was consistent with previous studies [15]. Lower-status residents lived in a poorer environment, such as more community violence, worse community environment, and higher crime rate, which would not be conducive to the formation of their community identity [33,34], and, thus, reduced their prosocial behavior. Moreover, Wang et al. [34] found community identity mediated the association between socioeconomic status and altruistic behavior, which provided indirect evidence for the mediating role of community identity between socioeconomic status and prosocial behavior. At the same time, the study also found that perceived control played a mediating role between socioeconomic status and prosocial behavior. Specifically, lower-status residents experienced lower perceived control, which in turn reduced their prosocial behavior. Lower-status residents were less educated, had less income and lower professional status, as well as a higher possibility of unemployment, leading to more uncertainty and unpredictability in their lives [38]. This would reduce perceived control and negatively affect their prosocial behavior.

In addition, this study also found that socioeconomic status affected prosocial behavior through the serially mediating role of community identity and perceived control. The group-based control restoration model pointed out that group identity could enhance individuals’ perceived control [40,41]. That is to say, it was difficult for lower-status adults to form a community identity, and the lack of community identity led to the reduction of their perceived control, which reduced their prosocial behavior.

There were some limitations in this study. First, the study used the questionnaire method to collect data, which cannot make causal inferences. Therefore, future studies should conduct experimental method or a longitudinal design to explore the causal relationships between variables. Second, the types of prosocial behavior are diverse, such as donation, altruism, and so on. Future studies could examine the effect of socioeconomic status on different types of prosocial behavior. Third, the object of prosocial behavior also affects whether individuals exhibit prosocial behavior. Future research should further distinguish whether the object of prosocial behavior comes from in-group members, strangers, and out-group members. Fourth, all the relationships, except for the one linking community identity and prosocial behavior, were weak. Future research can further explore the relationship between major variables by increasing the sample size. Finally, the present study only considered housing type, and future studies should further distinguish community characteristics.

Despite these limitations, the study also has some theoretical and practical implications. First, the current study extends cost–benefit analyses, the social cure perspective, and the group-based control restoration model in a community context. Second, the results showed that there was a positive correlation between socioeconomic status and prosocial behavior. Furthermore, community identity and perceived control played a serially mediating role in the link. Enhancing subjective status perceptions of lower-status individuals or adopting certain strategies to enhance their community identity and perceived control could improve their prosocial behavior.

## 5. Conclusions

In the present study, we examined the relationship between socioeconomic status and prosocial behavior, and the serially mediating role of community identity and perceived control in the connections among Chinese adults. The results indicated that socioeconomic status was positively associated with prosocial behavior. While, socioeconomic status affected prosocial behavior, not only through the independent mediating role of community identity or perceived control, but also through the serially mediating role of community identity and perceived control.

## Figures and Tables

**Figure 1 ijerph-18-10308-f001:**
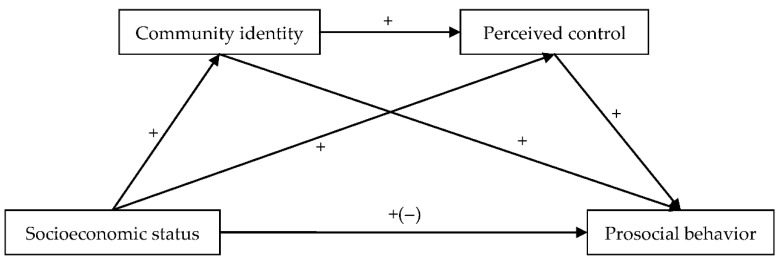
Study model.

**Figure 2 ijerph-18-10308-f002:**
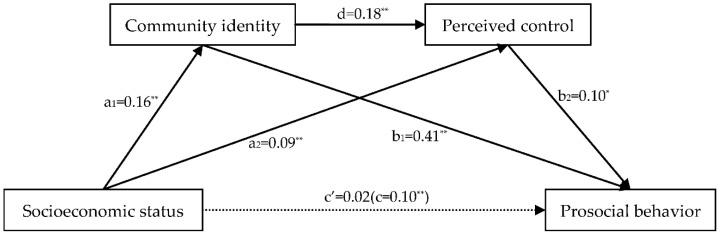
The pathway of the mediating model. ** *p* < 0.01 (two-tailed); * *p* < 0.05 (two-tailed).

**Table 1 ijerph-18-10308-t001:** Demographic characteristics of the sample.

Variables	Overall Sample (*N* = 477)	
	*N*	%
Gender		
Male	322	67.5
Female	155	32.5
Education level		
Primary school and below	2	0.4
Junior middle school	40	8.4
Senior high school	98	20.5
Associate degree	141	29.6
College graduate	176	36.9
Master and above	20	4.2
Occupation		
State and social managers	25	5.2
Managers	11	2.3
Private entrepreneurs	45	9.4
Professional and technical personnel	58	12.2
Office workers	40	8.4
Individual industrial and commercial households	59	12.4
Business and service employees	141	29.6
Industrial workers	26	5.5
Agricultural laborers	42	8.8
Urban and rural unemployed and semi-unemployed	30	6.3
Monthly income		
Less than 1000 yuan	10	2.1
1001–3000 yuan	64	13.4
3001–5000 yuan	162	34.0
5001–7000 yuan	127	26.6
7001–10,000 yuan	64	13.4
10,001–15,000 yuan	30	6.3
15,001–30,000 yuan	17	3.6
30,001–50,000 yuan	3	6.0
Housing type		
Owns a house	271	56.8
Without a house	206	43.2

**Table 2 ijerph-18-10308-t002:** Descriptive statistics, inter-correlations, composite reliability, average variance extracted, and fit indices of the major variables.

	CR	AVE	M ± SD	1	2	3	4
1. socieconomic status	0.795	0.659	0.00 ± 1.62	0.812			
2. community identity	0.944	0.679	4.02 ± 1.12	0.27 **	0.824		
3. perceived control	0.939	0.583	4.54 ± 0.99	0.20 **	0.22 **	0.764	
4. prosocial behavior	0.943	0.485	3.88 ± 0.59	0.18 **	0.43 **	0.19 **	0.696
Fit Indices	χ^2^/*df* = 2.96	*GFI* = 0.96	*IFI* = 0.97	*CFI* = 0.97	*RMSEA* = 0.06		

*Notes*: CR = composite reliability; AVE = average variance extracted; *χ*^2^/*df* = chi square/degrees of freedom; *GFI* = goodness-of-fit index; *CFI* = comparative fit index; *IFI* = incremental fit index; *RMSEA* = root mean square error of approximation. Square roots of average variances extracted are shown on the diagonal; the correlations between constructs are shown on the off-diagonal elements. ** *p* < 0.01 (two-tailed).

**Table 3 ijerph-18-10308-t003:** Test of the mediating effect.

	Effect	95% CI
Direct effect(c’)	0.02	(−0.03, 0.07)
Mediating effect 1 (a_1_ × b_1_)	0.07	(0.04, 0.09)
Mediating effect 2 (a_1_ × d × b_2_)	0.01	(0.01, 0.02)
Mediating effect 3 (a_2_ × b_2_)	0.01	(0.01, 0.02)
Total mediating effect	0.08	(0.05, 0.11)
Total effect (c)	0.10	(0.01, 0.15)

## Data Availability

The data presented in this study are available on request from the corresponding author. The data are not publicly available due to privacy.

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
