# Peer review of "Socioeconomic Status and Prosocial Behavior: The Mediating Roles of Community Identity and Perceived Control"

_ijerph, 2021, doi:10.3390/ijerph181910308_

Round 1
Reviewer 1 Report
This slighlty improved verions of the MS has still several shortcomings.
First, the authors did not manage to provide a source for the "MacArthur" scale.
Second, the rational for community identiy is more than vague. The auhturs failed to sufficiently explain their reasoning.
Finally, the authors did not apporprialty address my suggestion about differentiation between costs for helping and non-helping Furthermore, they argue the acting prosocial has its benefits. Thus, the authors way of arguing has remained inconclusive.
Author Response
Response to Reviewer 1 Comments
Dear Ariana Chen and the reviewer,
Thank you for your constructive comments on our paper (ijerph-1352605). We are very grateful to the editor and reviewers for the excellent level of detailed feedback offered to enable us to enhance the manuscript. We have carefully addressed the comments of the reviewers and highlighted(in red)the main changes made in the revised paper. Thank you for the opportunity to resubmit our manuscript for further consideration for publication in International Journal of Environmental Research and Public Health. All responses are made as follows.
Sincerely
Comments and Suggestions for Authors
This slighlty improved verions of the MS has still several shortcomings.
Point 1: First, the authors did not manage to provide a source for the "MacArthur" scale.
Response 1: Thank you for your constructive advice for improving our study. We have added a source for the "MacArthur" scale from line 210 to line 211.
“Subjective socioeconomic status was measured by the MacArthur Scale of Subjective Socioeconomic Status[50].”
References
- Adler, N.E.; Epel, E.S.; Castellazzo, G.; Ickovics, J.R. Relationship of subjective and objective social status with psychological and physiological functioning: Preliminary data in healthy, white women. Health Psychol. 2000, 19, 586–592.
Point 2: Second, the rational for community identiy is more than vague. The auhturs failed to sufficiently explain their reasoning.
Response 2: Thank you for your constructive advice for improving our study. We have added relevant sentence to make it more clearly from line 64 to line 67 and from line 98 to line 100.
“The Bronfenbrenner's ecosystem theory[17, 18] points out that a person's development is affected not only by personal characteristics, but also by microsystems. And socioeconomic status is a factor of the microsystem, and community identity is a personal characteristic variable.”
“Moreover, Wang et al.[34] found community identity made a mediating role between socioeconomic status and altruistic behavior.”
References
- Bronfenbrenner, U. The ecology of human development: Experiments by nature and design. Cambridge, MA: Harvard University Press. 1979.
- Bronfenbrenner, U. Recent advances in research on the ecology of human development. In R. K. Silbereisen, K. Eyferth, & G. Rudinger (Eds.), Development as action in context: Problem behavior and normal youth development (pp. 287–309). Heidelberg and New\brk: Springer-Verlag. 1986.
- Wang, Y.L.; Yang, C.; Hu, X,Y.; Chen, H. Community identity as a mediator of the relationship between socioeconomic status and altruistic behavior in Chinese residents. J Community App Soc Psychol. 2021, 1–11.
Point 3: Finally, the authors did not apporprialty address my suggestion about differentiation between costs for helping and non-helping Furthermore, they argue the acting prosocial has its benefits. Thus, the authors way of arguing has remained inconclusive.
Response 3: Thank you for your constructive advice for improving our study. We have added relevant sentences to make it conclusively from line 41 to line 44.
“cost-benefit analyses[12] showed that prosocial behavior consumed individuals’ resources, including time, effort, danger, embarrassment, and disruption of ongoing activities. With the increase of resources consumed, individuals would exhibit less prosocial behavior.”
References
- Dovidio, J.F.; Piliavin, J.A.; Schroeder, D.A.; Penner, L.A. The social psychology of prosocial behavior. Mahwah, NJ: Erlbaum, 2006.

Reviewer 2 Report
The authors have introduced the improvements indicated in the previous review. There is greater consistency in the formulation of the theoretical framework of the article by introducing by clarifications regarding some aspects that could be called into question, including improvements in the target group or population, the use of more precise bibliographic sources and the conceptualization of the concept of "socio-economic status".
However, and as was raised in the commentary that has been revised as "point 3", the way in which the term "socio-economic status" is conceptualized reveals a clear bias, as the studies are all focused in the direction of indicating that low socio-economic status is related to poorer living conditions in general, without taking into account that some positive factors also exist on "this side".
Again, it is recommended to consider revising this comment: “a more in-depth exposition of the information from these studies should be advisable, as the information is biased in favor of the interests of the researchers: are, all poor people lees caring? Is their contribution to the society less than of other people with higher status? The “other side of coin” must also be included, even if the aim is demonstrating that the socioeconomic status is beneficial for altruistic behavior or the pro-sociality”
The introduction of supplementary information in the analysis of the information is relevant and adds clarity to the document. Recommendations have been considered in the presentation of participants and the explanation of inclusion and exclusion criteria has been improved. The information provided regarding the socio-demographic characteristics of the participants should be presented in a clearer way (e.g. through a table) that is more visual and clearer. No justification has been given as to why there are differences by gender, years of experience and educational status, only the data has been reported. It is interesting that this is justified by contextualizing the information in the sample, as requested in the previous review.
The rest of the information requested (analysis of the data, explanation of variables, presentation of results) has been clearly improved. Even the discussion of the results has been improved, where greater precautions have been taken when estimating the associations between variables.
Author Response
Response to Reviewer 2 Comments
Dear Ariana Chen and the reviewer,
Thank you for your constructive comments on our paper (ijerph-1352605). We are very grateful to the editor and reviewers for the excellent level of detailed feedback offered to enable us to enhance the manuscript. We have carefully addressed the comments of the reviewers and highlighted(in red)the main changes made in the revised paper. Thank you for the opportunity to resubmit our manuscript for further consideration for publication in International Journal of Environmental Research and Public Health. All responses are made as follows.
Sincerely
Comments and Suggestions for Authors
The authors have introduced the improvements indicated in the previous review. There is greater consistency in the formulation of the theoretical framework of the article by introducing by clarifications regarding some aspects that could be called into question, including improvements in the target group or population, the use of more precise bibliographic sources and the conceptualization of the concept of "socio-economic status".
Point 1: However, and as was raised in the commentary that has been revised as "point 3", the way in which the term "socio-economic status" is conceptualized reveals a clear bias, as the studies are all focused in the direction of indicating that low socio-economic status is related to poorer living conditions in general, without taking into account that some positive factors also exist on "this side".
Response 1: We appreciate these suggestions. We have taken into account that some positive factors also exist on "this side" from line 38 to line 40 and from line 38 to line 40.
“For example, Low-status people were more likely to identify with social values oriented toward egalitarianism and the well-being of others[10], and showed higher levels of empathy[11].”
“Ample empirical studies found lower socioeconomic status was related to more prosocial behavior[10, 16].”
References
- Piff, P.K.; Kraus, M.W.; Côté, S.; Cheng, B.H.; Keltner, D. Having less, giving more: The influence of social class on prosocial behavior. J Pers Soc Psychol. 2010, 99(5), 771–784.
- van Kleef, G.A.; Oveis, C.; van der Löwe, I.; LuoKogan, A.; Goetz, J.; Keltner, D. Power, distress, and compassion: Turning a blind eye to the suffering of others. Psychol Sci. 2008, 19, 1315–1322.
- Greitemeyer, T.; Sagioglou, C. Does low (vs. high) subjective socioeconomic status increase both prosociality and aggression? Soc Psychol. 2018, 49(2), 76–87.
Point 2: Again, it is recommended to consider revising this comment: “a more in-depth exposition of the information from these studies should be advisable, as the information is biased in favor of the interests of the researchers: are, all poor people lees caring? Is their contribution to the society less than of other people with higher status? The “other side of coin” must also be included, even if the aim is demonstrating that the socioeconomic status is beneficial for altruistic behavior or the pro-sociality”
Response 2: We appreciate these suggestions. We have added the “other side of coin” from line 55 to line 63.
“On the other hand, from the perspective of social cognition of socioeconomic status[6], it pointed out that lower-status people had less material resources, faced more uncertainty and unpredictability, which led them to form a social cognitive tendency of contextualism. On the contrary, upper-status people had more social resources, which led them to form a social cognitive tendency of solipsism. Different cognitive tendencies had an effect on prosocial behavior. Ample empirical studies found lower socioeconomic status was related to more prosocial behavior[10, 16]. Overall, the relationship between socioeconomic status and prosocial behavior is still controversial.”
References
- Kraus, M.W.; Piff, P.K.; Mendoza-Denton, R.; Rheinschmidt, M.L.; Keltner, D. Social class, solipsism, and contextualism: How the rich are different from the poor. Psychol Rev. 2012, 119, 546–572.
- Piff, P.K.; Kraus, M.W.; Côté, S.; Cheng, B.H.; Keltner, D. Having less, giving more: The influence of social class on prosocial behavior. J Pers Soc Psychol. 2010, 99(5), 771–784.
- Greitemeyer, T.; Sagioglou, C. Does low (vs. high) subjective socioeconomic status increase both prosociality and aggression? Soc Psychol. 2018, 49(2), 76–87.
Point 3: The introduction of supplementary information in the analysis of the information is relevant and adds clarity to the document. Recommendations have been considered in the presentation of participants and the explanation of inclusion and exclusion criteria has been improved. The information provided regarding the socio-demographic characteristics of the participants should be presented in a clearer way (e.g. through a table) that is more visual and clearer. No justification has been given as to why there are differences by gender, years of experience and educational status, only the data has been reported. It is interesting that this is justified by contextualizing the information in the sample, as requested in the previous review.
Response 3: Thank you for your constructive advice for improving our study. We have presented the socio-demographic characteristics of the participants in a clearer way (e.g. through a table) that is more visual and clearer from line 149 to line 185.
Table 1. Demographic characteristics of sample.
|
Variables |
Overall sample (N = 477) |
|
|
N |
% |
|
|
Gender |
|
|
|
Male |
322 |
67.5 |
|
Female |
155 |
32.5 |
|
Education level |
|
|
|
Primary school and below |
2 |
0.4 |
|
Junior middle school |
40 |
8.4 |
|
Senior high school |
98 |
20.5 |
|
Associate degree |
141 |
29.6 |
|
College graduate |
176 |
36.9 |
|
Master and above |
20 |
4.2 |
|
Occupation |
|
|
|
State and social managers |
25 |
5.2 |
|
Managers |
11 |
2.3 |
|
Private entrepreneurs |
45 |
9.4 |
|
Professional and technical personnel |
58 |
12.2 |
|
Office workers |
40 |
8.4 |
|
Individual industrial and commercial households |
59 |
12.4 |
|
Business and service employees |
141 |
29.6 |
|
Industrial workers |
26 |
5.5 |
|
Agricultural laborers |
42 |
8.8 |
|
Urban and rural unemployed and semi-unemployed |
30 |
6.3 |
|
Monthly income |
|
|
|
Less than 1,000 yuan |
10 |
2.1 |
|
1,001~3,000 yuan |
64 |
13.4 |
|
3,001~5,000 yuan |
162 |
34.0 |
|
5,001~7,000 yuan |
127 |
26.6 |
|
7001~10,000 yuan |
64 |
13.4 |
|
10,001~15,000 yuan |
30 |
6.3 |
|
15,001~30,000 yuan |
17 |
3.6 |
|
30,001~50,000 yuan |
3 |
6.0 |
|
Housing type |
|
|
|
Own a house |
271 |
56.8 |
|
Without a house |
206 |
43.2 |
Point 4: The rest of the information requested (analysis of the data, explanation of variables, presentation of results) has been clearly improved. Even the discussion of the results has been improved, where greater precautions have been taken when estimating the associations between variables.
Response 4: Thank you again for your constructive advice for improving our study.

Reviewer 3 Report
This paper uses questionnaire data from 457 Chinese adults and structural equation models to examine whether community identity and perceived control mediate the relationship between socioeconomic status and prosocial behavior. I had a few suggestions I thought might help improve the paper:
--I found the models to be an interesting exercise, but the front end of the paper could do more to explain why these particular constructs should mediate the relationship between SES and prosocial behavior. As it reads now, the manuscript outlines why each item might be associate with prosocial behavior, but not why they should mediate the relationship between SES and social behavior. This important theoretical hinge would allow for a stronger argument for the contributions of the paper.
--perhaps related to the above, my initial reaction looking at the model was to wonder why the authors weren’t conceptualizing community identity and perceived control as moderators of the potential relationship between SES and prosocial behavior rather than mediators. The authors may have excellent explanations for this, and those explanations would likely come in the form of the increased attention to the theoretical linkages needed to argue for mediation mentioned above.
--in creating SES categories, there are some very small response cells. How did the authors manage this?
--I appreciated the detail concerning methods and found the fit measures to be appropriate.
Author Response
Response to Reviewer 3 Comments
Dear Ariana Chen and the reviewer,
Thank you for your constructive comments on our paper (ijerph-1352605). We are very grateful to the editor and reviewers for the excellent level of detailed feedback offered to enable us to enhance the manuscript. We have carefully addressed the comments of the reviewers and highlighted(in red)the main changes made in the revised paper. Thank you for the opportunity to resubmit our manuscript for further consideration for publication in International Journal of Environmental Research and Public Health. All responses are made as follows.
Sincerely
Comments and Suggestions for Authors
This paper uses questionnaire data from 457 Chinese adults and structural equation models to examine whether community identity and perceived control mediate the relationship between socioeconomic status and prosocial behavior. I had a few suggestions I thought might help improve the paper:
Point 1: I found the models to be an interesting exercise, but the front end of the paper could do more to explain why these particular constructs should mediate the relationship between SES and prosocial behavior. As it reads now, the manuscript outlines why each item might be associate with prosocial behavior, but not why they should mediate the relationship between SES and social behavior. This important theoretical hinge would allow for a stronger argument for the contributions of the paper.
Response 1: Thank you for your constructive advice for improving our study. We have added relevant sentence why they should mediate the relationship between socioeconomic status and social behavior from line 64 to line 67, from line 98 to line 100, from line 109 to line 111, and from line 116 to line 130.
“The Bronfenbrenner's ecosystem theory[17, 18] points out that a person's development is affected not only by personal characteristics, but also by microsystems. And socioeconomic status is a factor of the microsystem, and community identity is a personal characteristic variable.”
“Moreover, Wang et al.[34] found community identity made a mediating role between socioeconomic status and altruistic behavior.”
“Moreover, previous study[39] has found that control beliefs made a mediating role between socioeconomic status and exercise behavior or intention.”
“And Greenaway et al. [42] found that social identity would affect individual's mental health through perceived control. Social identity can also promote well-being by meeting the global psychological needs, including feeling of belonging, self-esteem, perceived control, and sense of meaning[43].
To sum up, although ample studies have identified the association between socio-economic status and prosocial behavior, they have not paid more attention to the underlying mechanism in the link. Here, we expanded previous studies in several aspects. First, although Wang et al. [34] indicated that community identity mediated the association between socioeconomic status and altruistic behavior, it is unclear whether community identity mediates the link of socioeconomic status with prosocial behavior. Second, no study has directly explored the mediating role of perceived control in the relation between socioeconomic status and prosocial behavior. Third, group-based control restoration model indicated that social identity would affected perceived control[40, 41], and perceived control mediated the association between social identity and mental health[42].”
References
- Bronfenbrenner, U. The ecology of human development: Experiments by nature and design. Cambridge, MA: Harvard Uni-versity Press. 1979.
- Bronfenbrenner, U. Recent advances in research on the ecology of human development. In R. K. Silbereisen, K. Eyferth, & G. Rudinger (Eds.), Development as action in context: Problem behavior and normal youth development (pp. 287–309). Heidelberg and New\brk: Springer-Verlag. 1986.
- Wang, Y.L.; Yang, C.; Hu, X,Y.; Chen, H. Community identity as a mediator of the relationship between socioeconomic status and altruistic behavior in Chinese residents. J Community App Soc Psychol. 2021, 1–11.
- Murray, T.C.; Rodgers, W.M.; Fraser, S. N. Exploring the relationship between socioeconomic status, control beliefs and exercise behavior: A multiple mediator model. J Behav Med. 2012, 35, 63–73.
- Fritsche, I.; Jonas, E.; Fankhänel, T. The role of control motivation in mortality salience effects on ingroup support and defense. J Pers Soc Psychol. 2008, 95, 524–541.
- Fritsche, I.; Jonas, E.; Ablasser, C.; Beyer, M.; Kuban, J.; Manger, A.; Schultz, M. The power of we: Evidence for group-based control. J Exp Soc Psychol. 2013, 49, 19–32.
- Greenaway, K.H.; Wright, R.; Willingham, J.; Reynolds, K.J.; Haslam, S.A. Shared identity is key to effective communication. Pers Soc Psychol Bull. 2015, 41, 171–182.
- Greenaway, K.H.; Cruwys, T.; Haslam, S.A.; Jetten, J. Social identities promote well-being because they satisfy global psychological needs. Eur J Soc Psychol. 2016, 46(3), 294–307.
Point 2: perhaps related to the above, my initial reaction looking at the model was to wonder why the authors weren’t conceptualizing community identity and perceived control as moderators of the potential relationship between SES and prosocial behavior rather than mediators. The authors may have excellent explanations for this, and those explanations would likely come in the form of the increased attention to the theoretical linkages needed to argue for mediation mentioned above.
Response 2: Thank you for your constructive advice for improving our study. First, previous study has shown that community identity mediated the relationship between socioeconomic status and altruistic behavior. This current study is a further expansion of previous research. Second, we examined the moderating role of perceived control in the relationship between socioeconomic status and prosocial behavior. Results showed that the moderating model was not valid. So, we conceptualized community identity and perceived control as mediators of the potential relationship between SES and prosocial behavior rather than moderators from line 121 to line 130.
“To sum up, although ample studies have identified the association between socioeconomic status and prosocial behavior, they have not paid more attention to the underlying mechanism in the link. Here, we expanded previous studies in several aspects. First, although Wang et al. [34] indicated that community identity mediated the association between socioeconomic status and altruistic behavior, it is unclear whether community identity mediates the link of socioeconomic status with prosocial behavior. Second, no study has directly explored the mediating role of perceived control in the relation between socioeconomic status and prosocial behavior. Third, group-based control restoration model indicated that social identity would affected perceived control[40, 41], and perceived control mediated the association between social identity and mental health[42].”
References
- Wang, Y.L.; Yang, C.; Hu, X,Y.; Chen, H. Community identity as a mediator of the relationship between socioeconomic status and altruistic behavior in Chinese residents. J Community App Soc Psychol. 2021, 1–11.
- Fritsche, I.; Jonas, E.; Fankhänel, T. The role of control motivation in mortality salience effects on ingroup support and defense. J Pers Soc Psychol. 2008, 95, 524–541.
- Fritsche, I.; Jonas, E.; Ablasser, C.; Beyer, M.; Kuban, J.; Manger, A.; Schultz, M. The power of we: Evidence for group-based control. J Exp Soc Psychol. 2013, 49, 19–32.
- Greenaway, K.H.; Wright, R.; Willingham, J.; Reynolds, K.J.; Haslam, S.A. Shared identity is key to effective communication. Pers Soc Psychol Bull. 2015, 41, 171–182.
Point 3: in creating SES categories, there are some very small response cells. How did the authors manage this?
Response 3: Thank you for your constructive advice for improving our study. We have added how we managed small response cells from line 192 to line 203.
“With reference to previous studies[34, 45], education level was divided into six categories: "primary school and below", "junior middle school", "senior high school", "associate degree", "college graduate", "master and above", which were assigned a score of 1-6, respectively. Occupation(reverse scoring) was divided into state and social managers, managers, private entrepreneurs, professional and technical personnel, office workers, individual industrial and commercial households, business and service employees, industrial workers, agricultural laborers, urban and rural unemployed and semiunemployed, which was based on cultural resources, economic resources, and organizational resources[46, 47]. Monthly income was divided into ten levels, including "less than 1,000 yuan", "1,001~3,000 yuan", "3,001~5,000 yuan", "5,001~7,000 yuan", "7001~10,000 yuan", "10,001~15,000 yuan", "15,001~30,000 yuan", "30,001~50,000 yuan", "50,001~100,000 yuan" and "more than 100,001 yuan"[48].”
References
- Wang, Y.L.; Yang, C.; Hu, X,Y.; Chen, H. Community identity as a mediator of the relationship between socioeconomic status and altruistic behavior in Chinese residents. J Community App Soc Psychol. 2021, 1–11.
- Christie, A.M.; Barling, J. Disentangling the indirect links between socioeconomic status and health: The dynamic roles of work stressors and personal control. J Appl Psychol. 2009, 94(6), 1466–1478.
- Lu, X.Y. The report on social stratification research in contemporary China. Beijing: Social Sciences Documentation Publishing House. 2002.
- Lu, X.Y. Social mobility in contemporary China.Beijing: Social sciences academic press(CHINA). 2004.
- Li, X.X.,; Ren, Z.H.; Hu, X.Y.; Guo, Y.Y. Why are undergraduates from lower-class families more likely to experience social anxiety? ——The multiple mediating effects of psychosocial resources and rejection sensitivity. J Psychol Sci. 2019, 42(6), 1354–1360.
Point 4: I appreciated the detail concerning methods and found the fit measures to be appropriate.
Response 4: Thank you again for your constructive advice for improving our study.

Reviewer 4 Report
This paper empirically examines the role of two underlying determinants of the association between socioeconomic status and prosocial behaviour: community identity and perceived control. The article is well-structured and the subject is interesting. The basic design and the statistical analysis are appropriate, but some revisions are necessary to enhance the scientific quality of the final article.
- Abstract. The authors should mention the sample selection method and the statistical analysis used. Also, they must highlight the main contributions.
- Introduction. It contains too little information and is not well documented/established.
a. The novelty/originality of the paper should be highlighted; what is new in this study that may benefit readers, what is the research gap of this paper or how it may contribute to existing literature or create new knowledge on this subject. It is not clear what are the main findings of the paper; i.e. the mediating roles of community identity and perceived control, separately, the serial mediating role or both?
There is also a question about how much added value this paper offers on top of the existing literature, especially for publication in a journal that specializes in environmental and health issues.
b. The link between socioeconomic status and prosocial behaviour is complex and multidimensional. In my opinion, it is not adequately presented here. For example, previous research has shown that the association may be positive or negative, depending on methodological grounds (survey, experiments, sample selection procedures, sample size, etc.). In fact, the authors denote by (+/-) such a controversial association in Figure 1, but they do not explain it. Moreover, Hypothesis 1 may be not adequately justified according to this link.
c. There are some confusing expressions along with the introduction and some references are not well inserted in the text; for example, the expression “the field experiment” (in line 41) refers to the experimental methodology in general or, by contrast, the authors make reference to a particular study which employs a field experiment. A similar issue happens in line 93 with “The Prisoner's Dilemma experiment”. Other misleading examples can be found when mentioning T1 and T2 (lines 81-84). I recommend the authors clarify and revise the writing for greater readability and understanding.
d. The research hypotheses should be justified along with the cited studies and they must be directly connected with Figure 1. Also, the statistical analysis used to explain Figure 1 should be described.
e. The main findings and the structure of the paper should be added at the end of the introduction.
- Material and Methods.
a. For further clarification, a descriptive table should be included with the information about participants: age, gender, education, income and occupation level. Likewise, the categories mentioned along the lines 147-156 result to be somehow repetitive, so they may be eliminated and replaced by some references to the added descriptive table.
b. The authors need to briefly describe the respondent selection method (how was the sample selected?) and the type of communities considered in the study to better understand the role of community and its connection with perceived control. In the introduction, the authors mention (line 32) that it is necessary to explore prosocial behaviour in the context of the community. However, a detailed description of the community characteristics is not provided.
b. The authors use a scale of eight items to measure community identity, but only two items are selected for the study. Is there any justification for that? I have the same concerns about perceived control and prosocial behaviour measures.
- Results section.
a. I don’t understand the correlation values corresponding to the main diagonal in Table 1. Shouldn’t they be equal to 1?
b. The relationships and coefficients (values and significance) shown in Figure 1 should be better explain. For example, what is the difference between c and c’ coefficients? Furthermore, though in the discussion the authors mention that some associations are weak, in the Results section there is no interpretation about that. Effect sizes shown in Table 2 should be also discussed.
c. In Table 2, regarding the mediating effect 2, it should be (a1*d*b2) instead of (a1*d*b1)?
d. The authors use two indicators (objective and subjective) for socioeconomic status. However, I didn’t find how these measures are considered in the statistical analysis. Are there some differences between them when explaining the association between socioeconomic status and prosocial behaviour?
- Discussion section.
a. It is simply descriptive and some repetitions can be found. There is no clear conclusion on why research findings are novel and what are their implications for the development of theory and policy design. I would invite authors to highlight their key contributions and add some discussions on "comparison" and "contrasting" with existing literature.
I hope my comments are useful and I wish the authors the best with the revision process.
Author Response
Response to Reviewer 4 Comments
Dear Ariana Chen and the reviewer,
Thank you for your constructive comments on our paper (ijerph-1352605). We are very grateful to the editor and reviewers for the excellent level of detailed feedback offered to enable us to enhance the manuscript. We have carefully addressed the comments of the reviewers and highlighted(in red)the main changes made in the revised paper. Thank you for the opportunity to resubmit our manuscript for further consideration for publication in International Journal of Environmental Research and Public Health. All responses are made as follows.
Sincerely
Comments and Suggestions for Authors
This paper empirically examines the role of two underlying determinants of the association between socioeconomic status and prosocial behaviour: community identity and perceived control. The article is well-structured and the subject is interesting. The basic design and the statistical analysis are appropriate, but some revisions are necessary to enhance the scientific quality of the final article.
Point 1: Abstract. The authors should mention the sample selection method and the statistical analysis used. Also, they must highlight the main contributions.
Response 1: Thank you for your constructive advice for improving our study. First, we have added the sample selection method and the statistical analysis used from line 11 to line 16. Second, we have highlighted the main contributions from line 23 to line 24.
“Using the convenient sampling technique, a total of 477 Chinese adults from Chinese communities, ranging in age from 20 to 65, finished the questionnaires for objective socioeconomic status, the MacArthur Scale of subjective socioeconomic status, eight-item community identity scale, perceived control scale, and prosocial tendencies measure. Bivariate correlation analysis and regression analysis were used to examine the relationships among the major variables.”
“have important implications for interventions aimed to improve prosocial behavior in lower-status individuals.”
Introduction. It contains too little information and is not well documented/established.
Point 2: The novelty/originality of the paper should be highlighted; what is new in this study that may benefit readers, what is the research gap of this paper or how it may contribute to existing literature or create new knowledge on this subject. It is not clear what are the main findings of the paper; i.e. the mediating roles of community identity and perceived control, separately, the serial mediating role or both?
Response 2: We appreciate these suggestions. First, we have highlighted the novelty/originality of the paper from line 121 to line 130. Second, we not only explored the mediating roles of community identity and perceived control, separately, but also the serial mediating role. We have modified the relevant expressions to make them more clearly from line 131 to line 139.
“To sum up, although ample studies have identified the association between socio-economic status and prosocial behavior, they have not paid more attention to the underlying mechanism in the link. Here, we expanded previous studies in several aspects. First, although Wang et al. [34] indicated that community identity mediated the association between socioeconomic status and altruistic behavior, it is unclear whether community identity mediates the link of socioeconomic status with prosocial behavior. Second, no study has directly explored the mediating role of perceived control in the relation between socioeconomic status and prosocial behavior. Third, group-based control restoration model indicated that social identity would affected perceived control[40, 41], and perceived control mediated the association between social identity and mental health[42].”
“Thus, based on theoretical and empirical studies, the present study aimed to explore the association between socioeconomic status and prosocial behavior, and the serial mediating role of community identity and perceived control in the association. Four hypotheses were proposed: (1)socioeconomic status was associated with prosocial behavior; (2)community identity mediated the association between socioeconomic status and prosocial behavior; (3)perceived control mediated the association between socioeconomic status; and (4) community identity and perceived control play a serial mediating role between socioeconomic status and prosocial behavior. The study model is presented in Figure 1.”
References
- Wang, Y.L.; Yang, C.; Hu, X,Y.; Chen, H. Community identity as a mediator of the relationship between socioeconomic status and altruistic behavior in Chinese residents. J Community App Soc Psychol. 2021, 1–11.
- Fritsche, I.; Jonas, E.; Fankhänel, T. The role of control motivation in mortality salience effects on ingroup support and defense. J Pers Soc Psychol. 2008, 95, 524–541.
- Fritsche, I.; Jonas, E.; Ablasser, C.; Beyer, M.; Kuban, J.; Manger, A.; Schultz, M. The power of we: Evidence for group-based control. J Exp Soc Psychol. 2013, 49, 19–32.
- Greenaway, K.H.; Wright, R.; Willingham, J.; Reynolds, K.J.; Haslam, S.A. Shared identity is key to effective communication. Pers Soc Psychol Bull. 2015, 41, 171–182.
Point 3: There is also a question about how much added value this paper offers on top of the existing literature, especially for publication in a journal that specializes in environmental and health issues.
Response 3: We appreciate these suggestions. First, we emphasized that prosocial behavior was closely related to individuals’ mental health from line 32 to line 33. Therefore, it is of great significance to explore how to improve individuals’ prosocial behavior. Second, previous studies related to prosocial behavior have been published on International Journal of Environmental Research and Public Health. Luo et al. (2021) explored the mediating role of gratitude and the moderating role of the school climate between parental warmth and prosocial behavior. Weiß, Hein, and Hewig(2021) were interested in whether facial expressions of the recipient in the dictator game influence dictators’ behavior. Thus, we think that it provides added value for publication in a journal that specializes in environmental and health issues and is appropriate.
“It can improve individuals’ happiness[2, 3], perceptions of meaning in life[4], and relieve depression, anxiety[5].”
References
Luo, H., Liu, Q., Yu, C., & Nie, Y. (2021). Parental warmth, gratitude, and prosocial behavior among chinese adolescents: the moderating effect of school climate. International Journal of Environmental Research and Public Health, 18(13), 7033.
Weiß, M., Hein, G., & Hewig, J. (2021). Between joy and sympathy: Smiling and sad recipient faces increase prosocial behavior in the dictator game. International Journal of Environmental Research and Public Health, 18(11), 6172.
- Lawton, R.N.; Gramatki, I.; Watt, W.; Fujiwara, D. Does volunteering make us happier, or are happier people more likely to volunteer? Addressing the problem of reverse causality when estimating the well-being impacts of volunteering. J Happiness Stud. 2020, 22, 1–26.
- Park, S.Q.; Kahnt, T.; Dogan, A.; Strang, S.; Fehr, E.; Tobler, P.N. A neural link between generosity and happiness. Nat Commun. 2017, 8, 15964.
- Klein, N. (2017). Prosocial behavior increases perceptions of meaning in life. J Posit Psychol. 2017, 12, 354–361.
- Tsuboi, H.; Hirai, H.; Kondo, K. Giving social support to outside family may be a desirable buffer against depressive symptoms in community-dwelling older adults: Japan gerontological evaluation study. Bio Psycho Soc Med. 2016, 10, 18.
Point 4: The link between socioeconomic status and prosocial behaviour is complex and multidimensional. In my opinion, it is not adequately presented here. For example, previous research has shown that the association may be positive or negative, depending on methodological grounds (survey, experiments, sample selection procedures, sample size, etc.). In fact, the authors denote by (+/-) such a controversial association in Figure 1, but they do not explain it. Moreover, Hypothesis 1 may be not adequately justified according to this link.
Response 4: Thank you for your constructive advice for improving our study. Owing to the debates and mixed evidence, we revised our hypotheses presented as exploratory from line 133 to line 141.
“Four hypotheses were proposed: (1)socioeconomic status was associated with prosocial behavior; (2) community identity mediated the association between socioeconomic status and prosocial behavior; (3)perceived control mediated the association between socioeconomic status; and (4) community identity and perceived control play a serial mediating role between socioeconomic status and prosocial behavior. The study model is presented in Figure 1.”
Figure 1. Study model.
Point 5: There are some confusing expressions along with the introduction and some references are not well inserted in the text; for example, the expression “the field experiment” (in line 41) refers to the experimental methodology in general or, by contrast, the authors make reference to a particular study which employs a field experiment. A similar issue happens in line 93 with “The Prisoner's Dilemma experiment”. Other misleading examples can be found when mentioning T1 and T2 (lines 81-84). I recommend the authors clarify and revise the writing for greater readability and understanding.
Response 5: Thank you for your constructive advice for improving our study. We have clarified and revised the writing for greater readability and understanding from line 47 to line 49 and from line 89 to line 93.
“Andreoni, Nikiforakis, and Stoop[14] found that the poor were less likely to return misdelivered envelopes, even if it was not good for them to keep them.”
“Researchers[32] conducted a two-wave longitudinal online survey studied during the COVID-19 pandemic, and found volunteer role identity at T1(pre-pandemic) positively affected perceptions of volunteer-beneficiary intergroup closeness at T1, which also in turn affected community identity at T1. This, in turn, positively predicted COVID-19 collaborative assistance for T2 (3 months later).”
References
- Andreoni, J.; Nikiforakis, N.; Stoop, J. Higher socioeconomic status does not predict decreased prosocial behavior in a field experiment. Nat Commun. 2021, 12, 4266.
- Wakefield, J.R.; Bowe, M.; Kellezi, B. Who helps and why? A longitudinal exploration of volunteer role identity, intergroup closeness, and community identification as predictors of coordinated helping during the COVID-19 pandemic. PsyArXiv. Retrieved from https://doi.org/10.31234/osf.io/8kcyj, 2021.
Point 6: The research hypotheses should be justified along with the cited studies and they must be directly connected with Figure 1. Also, the statistical analysis used to explain Figure 1 should be described.
Response 6: We appreciate these suggestions. First, we have revised relevant sentence from line 133 to line 141. Second, we have added the statistical analysis used to explain Figure 1 should be described from line 254 to line 258.
“Four hypotheses were proposed: (1)socioeconomic status was associated with prosocial behavior; (2) community identity mediated the association between socioeconomic status and prosocial behavior; (3)perceived control mediated the association between socioeconomic status; and (4) community identity and perceived control play a serial mediating role between socioeconomic status and prosocial behavior. The study model is presented in Figure 1.”
Figure 1. Study model.
“Third, bivariate correlation analysis was used to examine the relationships among the major variables. Fourth, we tested the mediated model using the PROCESS macro (http://www.afhayes.com) for SPSS[60], with 5,000 iterations of computing samples and bias-corrected 95% confidence intervals. If the confidence interval of 95% does not include zero, the result was significant at the p<0.05 level.”
References
- Hayes, A.F. Process: A versatile computational tool for observed variable mediation, moderation, and conditional process modeling[Whitepaper]. Retrieved from http://www.afhayes.com/public/ process2012.pdf, 2012.
Point 7: The main findings and the structure of the paper should be added at the end of the introduction.
Response 7: Thank you for your constructive advice for improving our study. We have added the hypotheses at the end of the introduction for further verification from line 121 to line 141.
“To sum up, although ample studies have identified the association between socioeconomic status and prosocial behavior, they have not paid more attention to the underlying mechanism in the link. Here, we expanded previous studies in several aspects. First, although Wang et al. [34] indicated that community identity mediated the association between socioeconomic status and altruistic behavior, it is unclear whether community identity mediates the link of socioeconomic status with prosocial behavior. Second, no study has directly explored the mediating role of perceived control in the relation between socioeconomic status and prosocial behavior. Third, group-based control restoration model indicated that social identity would affected perceived control[40, 41], and perceived control mediated the association between social identity and mental health[42]. Thus, based on theoretical and empirical studies, the present study aimed to explore the association between socioeconomic status and prosocial behavior, and the serial mediating role of community identity and perceived control in the association. Four hypotheses were proposed: (1)socioeconomic status was associated with prosocial behavior; (2) community identity mediated the association between socioeconomic status and prosocial behavior; (3)perceived control mediated the association between socioeconomic status; and (4) community identity and perceived control play a serial mediating role between socioeconomic status and prosocial behavior. The study model is presented in Figure 1.”
”
Figure 1. Study model.
References
- Wang, Y.L.; Yang, C.; Hu, X,Y.; Chen, H. Community identity as a mediator of the relationship between socioeconomic status and altruistic behavior in Chinese residents. J Community App Soc Psychol. 2021, 1–11.
- Fritsche, I.; Jonas, E.; Fankhänel, T. The role of control motivation in mortality salience effects on ingroup support and defense. J Pers Soc Psychol. 2008, 95, 524–541.
- Fritsche, I.; Jonas, E.; Ablasser, C.; Beyer, M.; Kuban, J.; Manger, A.; Schultz, M. The power of we: Evidence for group-based control. J Exp Soc Psychol. 2013, 49, 19–32.
- Greenaway, K.H.; Wright, R.; Willingham, J.; Reynolds, K.J.; Haslam, S.A. Shared identity is key to effective communication. Pers Soc Psychol Bull. 2015, 41, 171–182.
Material and Methods.
Point 8: For further clarification, a descriptive table should be included with the information about participants: age, gender, education, income and occupation level. Likewise, the categories mentioned along the lines 147-156 result to be somehow repetitive, so they may be eliminated and replaced by some references to the added descriptive table.
Response 8: Thank you for your constructive advice for improving our study. We have added a descriptive table from line 149 to line 185.
Table 1. Demographic characteristics of sample.
|
Variables |
Overall sample (N = 477) |
|
|
N |
% |
|
|
Gender |
|
|
|
Male |
322 |
67.5 |
|
Female |
155 |
32.5 |
|
Education level |
|
|
|
Primary school and below |
2 |
0.4 |
|
Junior middle school |
40 |
8.4 |
|
Senior high school |
98 |
20.5 |
|
Associate degree |
141 |
29.6 |
|
College graduate |
176 |
36.9 |
|
Master and above |
20 |
4.2 |
|
Occupation |
|
|
|
State and social managers |
25 |
5.2 |
|
Managers |
11 |
2.3 |
|
Private entrepreneurs |
45 |
9.4 |
|
Professional and technical personnel |
58 |
12.2 |
|
Office workers |
40 |
8.4 |
|
Individual industrial and commercial households |
59 |
12.4 |
|
Business and service employees |
141 |
29.6 |
|
Industrial workers |
26 |
5.5 |
|
Agricultural laborers |
42 |
8.8 |
|
Urban and rural unemployed and semi-unemployed |
30 |
6.3 |
|
Monthly income |
|
|
|
Less than 1,000 yuan |
10 |
2.1 |
|
1,001~3,000 yuan |
64 |
13.4 |
|
3,001~5,000 yuan |
162 |
34.0 |
|
5,001~7,000 yuan |
127 |
26.6 |
|
7001~10,000 yuan |
64 |
13.4 |
|
10,001~15,000 yuan |
30 |
6.3 |
|
15,001~30,000 yuan |
17 |
3.6 |
|
30,001~50,000 yuan |
3 |
6.0 |
|
Housing type |
|
|
|
Own a house |
271 |
56.8 |
|
Without a house |
206 |
43.2 |
Point 9: The authors need to briefly describe the respondent selection method (how was the sample selected?) and the type of communities considered in the study to better understand the role of community and its connection with perceived control. In the introduction, the authors mention (line 32) that it is necessary to explore prosocial behaviour in the context of the community. However, a detailed description of the community characteristics is not provided.
Response 9: We appreciate these suggestions. First, we have briefly described the respondent selection method(line 144). Second, we have added the sentences from line 380 to line 381. This is a good suggestion. We only consider housing type in the study, and future studies should further distinguish community characteristics.
“Using the convenient sampling technique”
“Finally, the present study only considered housing type, and future studies should further distinguish community characteristics. ”
Point 10: The authors use a scale of eight items to measure community identity, but only two items are selected for the study. Is there any justification for that? I have the same concerns about perceived control and prosocial behaviour measures.
Response 10: Thank you for your constructive advice for improving our study. We used complete scales to measure community identity, perceived control, and prosocial behavior, and only enumerated some items in the full text.
Results section.
Point 11: I don’t understand the correlation values corresponding to the main diagonal in Table 1. Shouldn’t they be equal to 1?
Response 11: Thank you for your constructive advice for improving our study. The main diagonal corresponds not to the correlation values, but to the square roots of average variances extracted. We have added relevant sentence to make it more clearly from line 298 to line 299.
“Square roots of average variances extracted was showed on the diagonal”
Point 12: The relationships and coefficients (values and significance) shown in Figure 1 should be better explain. For example, what is the difference between c and c’ coefficients? Furthermore, though in the discussion the authors mention that some associations are weak, in the Results section there is no interpretation about that. Effect sizes shown in Table 2 should be also discussed.
Response 12: Thank you for your constructive advice for improving our study. First, we have added relevant sentence to make it more clearly from line 324 to line 331. Second, in order to better distinguish Results from Discussion, we only present results objectively in the Results section.
Table 3. Test of the mediating effect
|
|
Effect |
95%CI |
|
direct effect(c’) |
0.02 |
[-0.03, 0.07] |
|
Mediating effect 1(a1*b1) |
0.07 |
[0.04, 0.09] |
|
Mediating effect 2(a1*d*b2) |
0.01 |
[0.01, 0.02] |
|
Mediating effect 3(a2*b2) |
0.01 |
[0.01, 0.02] |
|
Total mediating effect |
0.08 |
[0.05, 0.11] |
|
Total effect(c) |
0.10 |
[0.01, 0.15] |
Point 13: In Table 2, regarding the mediating effect 2, it should be (a1*d*b2) instead of (a1*d*b1)?
Response 13: Thank you for your constructive advice for improving our study. We have made revision.
Point 14: The authors use two indicators (objective and subjective) for socioeconomic status. However, I didn’t find how these measures are considered in the statistical analysis. Are there some differences between them when explaining the association between socioeconomic status and prosocial behaviour?
Response 14: Thank you for your constructive advice for improving our study. We have added relevant sentence to make it more clearly from line 189 to line 190.
“According to the existing research[44], socioeconomic status was calculated through adding the standard scores of subjective and objective indicators.”
References
- Tan, J.J.X.; Kraus, M.W. Lay theories about social class buffer lower-class individuals against poor self-rated health and negative affect. Pers Soc Psychol Bull. 2015, 41, 446–461.
Discussion section.
Point 15: It is simply descriptive and some repetitions can be found. There is no clear conclusion on why research findings are novel and what are their implications for the development of theory and policy design. I would invite authors to highlight their key contributions and add some discussions on "comparison" and "contrasting" with existing literature.
Response 15: Thank you for your constructive advice for improving our study. First, we have added reasons why the present study was novel from line 121 to line 130. Second, we have added relevant sentences to highlight their theoretical and practical implications from line 382 to line 389.
“To sum up, although ample studies have identified the association between socio-economic status and prosocial behavior, they have not paid more attention to the underlying mechanism in the link. Here, we expanded previous studies in several aspects. First, although Wang et al. [34] indicated that community identity mediated the association between socioeconomic status and altruistic behavior, it is unclear whether community identity mediates the link of socioeconomic status with prosocial behavior. Second, no study has directly explored the mediating role of perceived control in the relation between socioeconomic status and prosocial behavior. Third, group-based control restoration model indicated that social identity would affected perceived control[40, 41], and perceived control mediated the association between social identity and mental health[42].”
“Despite these limitations, the study also has some theoretical and practical implications. First, the current study extends cost-benefit analyses, the social cure perspective, and group-based control restoration model in community context. Second, result showed that there was a positive correlation between socioeconomic status and prosocial behavior. And, community identity and perceived control played a serial mediating role in the link. Enhancing subjective status perception of lower-status individuals or adopting certain strategies to enhance their community identity and perceived control can improve their prosocial behavior.”
References
- Wang, Y.L.; Yang, C.; Hu, X,Y.; Chen, H. Community identity as a mediator of the relationship between socioeconomic status and altruistic behavior in Chinese residents. J Community App Soc Psychol. 2021, 1–11.
- Fritsche, I.; Jonas, E.; Fankhänel, T. The role of control motivation in mortality salience effects on ingroup support and defense. J Pers Soc Psychol. 2008, 95, 524–541.
- Fritsche, I.; Jonas, E.; Ablasser, C.; Beyer, M.; Kuban, J.; Manger, A.; Schultz, M. The power of we: Evidence for group-based control. J Exp Soc Psychol. 2013, 49, 19–32.
- Greenaway, K.H.; Wright, R.; Willingham, J.; Reynolds, K.J.; Haslam, S.A. Shared identity is key to effective communication. Pers Soc Psychol Bull. 2015, 41, 171–182.
Point 16: I hope my comments are useful and I wish the authors the best with the revision process.
Response 16: Thank you again for your constructive advice for improving our study.

Round 2
Reviewer 1 Report
I'm still not convinced why communities are a special type of social categories. Moreover, it is still unclear whether the alleged measures of identification measures identification.
Author Response
Response to Reviewer 1 Comments
Dear Ariana Chen and the reviewer,
Thank you for your constructive comments on our paper (ijerph-1352605). We are very grateful to the editor and reviewers for the excellent level of detailed feedback offered to enable us to enhance the manuscript. We have carefully addressed the comments of the reviewers and highlighted(in red)the main changes made in the revised paper. Thank you for the opportunity to resubmit our manuscript for further consideration for publication in International Journal of Environmental Research and Public Health. All responses are made as follows.
Sincerely
Comments and Suggestions for Authors
I'm still not convinced why communities are a special type of social categories. Moreover, it is still unclear whether the alleged measures of identification measures identification.
Response 1: Thank you for your constructive advice for improving our study. First, Tönnies(1912) proposed a dichotomous distinction between society and community as he defined Gemeinschaft (community) as intimate, private, and exclusive living together, whereas the larger Gesellschaft (society) was seen as the public life, that is, the world itself. Karp, Stone and Yoels (1977) identified three elements defining communities: sustained social interaction, shared attributes and values, and a delineated geographical space. So, community is only a part of society or a microcosm. Second, the Community Identity Scale has been widely used to measure community identity. For example, Yang et al., (2020) explored the mediating effect of community identity on the association between sense of community responsibility and altruistic behavior. Wang et al., (2021) explored the mediating role of community identity in the link between socioeconomic status and altruistic behavior. Yang and Xin(2016) examined how time pressure and community identity affected urban residents’ ingroup emergency helping intention. Wang et al., (2021) addressed whether heterogenous community identity profiles emerged and how they differed in coronavirus disease 2019 (COVID-19)-related community participation. Therefore, we think that the Community Identity Scale can well measure community identity.
References
Karp, D., Stone, G., & Yoels, W. (1977). Being urban: A social psychological view of city life. Lexington, MA: Heath and Company.
Tönnies, F. (1912). Gemeinschaft und Gesellschaft: Grundbegriffe der reinen Soziologie, 2nd edition, Berlin: Curtius.
Wang, X., Yang, Z., Xin, Z., Wu, Y., & Qi, S. (2021). Community identity profiles and COVID-19-related community participation. Journal of Community & Applied Social Psychology, 1–13.
Wang, Y., Yang, C., Hu, X., & Chen, H. (2021). Community identity as a mediator of the relationship between socioeconomic status and altruistic behaviour in Chinese residents. Journal of Community & Applied Social Psychology, 31, 26–38.
Yang, C., Wang, Y., Hall, B. J., & Chen, H. (2020). Sense of community responsibility and altruistic behavior in Chinese community residents: The mediating role of community identity. Current Psychology, 1–11.
Yang, Z., & Xin, Z. Q. (2016). Community identity increases urban residents' in-group emergency helping intention. Journal of Community & Applied Social Psychology, 26(6), 467–480.
